


# An exploratory performance assessment of the *CHIMERE* model (version 2017r4) for the northwestern Iberian Peninsula and the summer season

Swen Brands[1,2], Guillermo Fernández-García[1], Marta García Vivanco[3], Marcos Tesouro Montecelo[1], Nuria Gallego Fernández[4], Anthony David Saunders Estévez[2,4], Pablo Enrique Carracedo García[1], Anabela Neto Venancio[1,2], Pedro Melo Da Costa[1,2], Paula Costa Tomé[2,4], Cristina Otero[2,4], María Luz Macho[1], and Juan Taboada[1,2]

[1]MeteoGalicia - Consellería de Medio Ambiente, Territorio e Vivenda, Xunta de Galicia, Santiago de Compostela, Spain
[2]Tragsatec, Santiago de Compostela, Spain
[3]Centro de Investigaciones Energéticas, Medioambientables y Tecnológicas (CIEMAT), Madrid, Spain
[4]Servicio de Calidad del Aire - Consellería de Medio Ambiente, Territorio e Vivenda, Xunta de Galicia, Santiago de Compostela, Spain

**Correspondence:** Swen Brands (swen.brands@gmail.com)

**Abstract.** Here, the capability of the chemical weather forecasting model CHIMERE (version 2017r4) to reproduce surface ozone, particulate matter and nitrogen dioxide concentrations in complex terrain is investigated for the period from June 21 to August 21, 2018. The study area is the northwestern Iberian Peninsula, where both coastal and mountain climates can be found in direct vicinity and a large fraction of the land area is covered by forests. Driven by lateral boundary conditions

5   from the ECMWF Composition Integrated Forecast System, anthropogenic emissions from two commonly used top-down inventories and meteorological data from the Weather Research and Forecasting Model, CHIMERE's performance with respect to observations is tested with a range of sensitivity experiments. We assess the effects of 1) an increase in horizontal resolution, 2) an increase in vertical resolution, 3) the use of distinct model chemistries and 4) the use of distinct anthropogenic emissions inventories, downscaling techniques and landuse databases. In comparsion with the older HTAP emission inventory downscaled

10  with basic options, the updated and sophistically downscaled EMEP inventory only leads to partial model improvements and so does the computationally costly horizontal resolution increase. Model performance changes caused by the choice of distinct chemical mechanisms are not systematic either and rather depend on the considered anthropgenic emission configuration and pollutant. Albeit the results are thus heterogeneous in general terms, the model's response to a *vertical* resolution increase confined to the lower to middle troposphere is homogeneous in the sense of improving virtually all verification aspects. We

15  conclude that, as long as the aforementioned top-down emission inventories are used, it is generally not necessary to use a horizontal model mesh much finer than the native grid of the inventories. A relatively coarse horizontal mesh combined with 20 model layers between 999 and 500 hPa is sufficient to yield balanced results. The chemical mechanism should be chosen as a function of the intended application.





*Copyright statement.* TEXT

## 1 Introduction

Motivated by the air quality legislation of the European Union (EU, 2008), many governmental air quality departments are currently demanding air quality forecasting schemes based on numerical models (Thunis et al., 2016), and the need for accurate and computationally efficient predictions in this field is perhaps greatest than ever before. For Europe as a whole, the most important real-time prediction system available to date is provided by the Copernicus Atmosphere Monitoring Service (Marécal et al., 2015), comprising an ensemble of currently seven chemical weather forecasting (CWF) models[1] run for the entire continent at a horizontal resolution of $0.1°$. to $0.25°$ in longitude and $0.1°$ to $0.2°$ in latitude. In addition to this short-term prediction system, several large research initiatives have been issued during the last two decades in order to assess the *climatological* properties of atmospheric composition, including the detection of long-term trends resulting from emission reductions induced by the Convention on Long-range Transboundary Air Pollution (CLRTAP, 2019). The final aim of these efforts is to find model configurations, or ensembles thereof, that can be used as surrogates for real observations in order to assess whether emission reductions actually have lead, or would lead, to changes in the atmosphere's composition on climatological time-scales (Vautard et al., 2006; Jonson et al., 2006; Colette et al., 2011; Wilson et al., 2012; Banzhaf et al., 2015; Colette et al., 2017; Im et al., 2018b, a; Vivanco et al., 2018; Theobald et al., 2019).

Complementary to these large-scale efforts, usually conducted with a single configuration of a given model (Bessagnet et al., 2016), small-scale sensitivity tests for particular models are still relevant since they can be run with more sophisticated model configurations than their large-scale counterparts and are therefore more interesting for regional prediction systems, such as those demanded by national or regional governments (Banzhaf et al., 2012; Beegum et al., 2016; Flamant et al., 2018). Further, following the concept of seamless prediction (Palmer et al., 2008), lessons learned from short-term prediction systems for relatively small geographical areas might as well be important for longer lead-times and larger areas.

Previous sensitivity studies have identified several *factors* influencing the models' capability to correctly reproduce observed values, hereafter referred to as "model performance" (Giorgi and Francisco, 2000; Chang and Hanna, 2004). Among these factors, the meteorological model used to drive the chemical model and the accuracy of the underlying emission datasets play a key role and have been assessed in a number of studies (Menut, 2008; Markakis et al., 2015; Colette et al., 2017; Otero et al., 2018; Vivanco et al., 2018). The resolution of the model mesh used to discretize the chemical reactions and atmospheric dynamics is also important and, when it is increased, a trade-off between potential performance gains and computational cost must been made in practice. In what concerns the *horizontal* resolution, performance gains have been reported up to a scale of approximately 12 km for a number of models, such as WRF-CHEM and CHIMERE (Valari and Menut, 2008; Schaap et al., 2015; Crippa et al., 2017). However, a further resolution increase does not guarantee further performance gains. Namely, beyond the 12 km threshold, Misenis and Zhang (2010) reported heterogeneous results for WRF-CHEM that strongly depend on the considered time period. For the use of CHIMERE and focussing on surface $O_3$ concentrations, Valari and Menut (2008)

---

[1] see Kukkonen et al. (2012) for an overview of these models



even found a performance *loss* which they attributed to a noise increase in the emission fluxes and meteorological input data at higher resolutions. Regarding the role of *vertical* resolution, an increase therein has been found to improve the modelled particulate matter (PM) concentrations during desert dust events when using WRF-CHEM (Teixeira et al., 2016). CHIMERE's performance, however, was found to be only weakly affected by this kind of resolution increase (Menut et al., 2013a; Markakis
et al., 2015).

Representing the number and complexity of the considered chemical reactions, several *chemistry mechanisms* are usually available for a given model and switching from one mechanism to another can also affect the model's performance (Balzarini et al., 2015; Karlický et al., 2017). In recent CHIMERE versions, the SAPRC-07A mechanism (hereafter: SAPRC) has been included as an alternative to the full or reduced versions of the Melchior mechanism (Carter, 2010; Mailler et al., 2017) but, to
the authors' knowledge, related sensitivity tests are sparse to date.

A common limitation of small-scale sensitivity studies is that their conclusions, strictly speaking, only hold for the considered region, time period or season of the year. In this context, most of the aforementioned conclusions for CHIMERE (the model applied here) have been drawn for the *Île de France* region, which is densely populated, relatively flat and not directly influenced by sea-salt emissions. The model has been applied for a number of other regions but the map is still incomplete and
sensitivity testing is not the main focus of the corresponding studies (Mazzeo et al., 2018; Menut et al., 2018; Monteiro et al., 2018; Brasseur et al., 2019; Deroubaix et al., 2019).

This is where the present study comes into play: For the two month period from June 21 to August 21, 2018 a series of 19 sensitivity tests has been run with CHIMERE over the *northwestern Iberian Peninsula*, a region characterized by forested mountain terrain, a complex coastline and the advection of sea-salt from the surrounding Atlantic Ocean that is quite different
from the *Île de France* region. The applied tests will quantify the effects arising from 1) an increase in model resolution (vertical and/or horzizontal), 2) switching from one chemistry mechanims to another (full Melchior or SAPRC in this case) and 3) changing the applied anthropgenic emissions inventory, downscaling strategy and landuse database. To this end, version 2017r4 of the CHIMERE model is used (Mailler et al., 2017) in combination with the HTAP v2.2 and EMEP emission inventories of the years 2010 and 2017 respectively (Janssens-Maenhout et al., 2015; EMEP/CEIP, 2019). Long-range transport of e.g. ozone
and its precursors or Saharan dust are *not* accounted for by running CHIMERE on a large domain covering all relevant remote emission sources (Bessagnet et al., 2017; Pay et al., 2019), but by using a far smaller domain ingesting the global forecasts provided by the European Centre for Medium-Range Weather Forecasts (ECMWF) Composition Integrated Forecasting system (C-IFS) at its lateral boundaries (Flemming et al., 2015). This strategy largely reduces the computational costs and is an interesting alternative to simulating long-range transport phenomena with the CHIMERE model itself (Bessagnet et al., 2017).
In Section 2, the applied data, model configurations and verification measures are described. Results are presented in Section 3 and a discussion and some general conclusions are provided in Section 4.





## 2 Data and Methods

In this section, the meteorological input data and general characteristics of the CHIMERE experiments are depicted first
(Section 2.1), followed by a description of the two applied emission inventories (Section 2.2) and individual model experiments

(Section 2.3). The in-situ station network used as reference for verification is introduced in Section 2.4. The section closes with
a description of the verfication measures used to estimate CHIMERE's performance for the applied experiments (see Section
2.5).

### 2.1 Meteorological Input and General Characteristics of the CHIMERE Experiments

The meteorological input data for the CHIMERE experiments is provided by the Weather Research and Forecasting (WRF)

model version 3.5 (Skamarock et al., 2008), driven by Global Forecast System (GFS) forecasts initialized at 00 UTC (Caplan
et al., 1997). WRF is run on three domains, a continental-scale domain having a resolution of 36km, followed by a regional
domain covering southwestern Europe at a resolution of 12km and, finally, a 4km domain covering our study region, the
northwestern Iberian Peninsula. For these domains, WRF is executed with a minimum time step of 216, 72 and 24 seconds
and a maximum time step of 360, 180 and 60 seconds, respectively. All domains comprise 33 vertical layers with a model

top at 10 hPa. A detailed overview of the WRF physics can be found in Table 1. In this configuration, WRF has been run for
now more than a decade at the meteorological office of the Galician government (MeteoGalicia) in order to provide real-time
meterological foreasts for the northwestern Iberian Peninsula. It is able to simulate the orographic and coastal effects on the
local weather reasonably well, which is illustrated in supplementary Figure 1 for a typical summertime heat day (August 5th,
2018).

With this meteorological input, version 2017r4 of the CHIMERE model is run on two domains: a coarse domain having a
horizontal resolution of $0.15° \times 0.15°$ ($longitude \times latitude$), and a fine domain, nested into the former, having a resolution
of $0.05° \times 0.04°$ (see Figure 1a). Note that the terms "coarse" and "fine" shall hereafter refer to the CHIMERE domains,
not the WRF domains, if not otherwise stated. Biogenic emissions comprising VOCs and NO are from the MEGAN model
version 2.04 (Guenther et al., 2006) and mineral dust emissions within the CHIMERE domains are calculated on the basis

of the United States Geological Survey landuse dataset (Loveland et al., 2000). The Alfaro and Gomes (2001) saltation and
sandblasting scheme, optimized by Menut et al. (2005), and the surface wind threshold described in Shao and Lu (2000)
are used throughout all experiments. The effect of soil moisture on dust emissions (Fécan et al., 1998) is activated and so
are sea-salt emissions. Vertical advection is achieved by the upwind scheme, horizontal advection by the more complex van
Leer (1979) scheme. Carbonaceous species as well as the interaction between aerosols and gases are taken into account by

the model and the number of Gauss-Seidel iterations is set to 3 because the model occasionally develops unrealistic waves
with lower numbers. Wind speed reduction in urban areas (the so called "urban correction") is deactivated, and so is the
resuspension process. A complete list of the internal CHIMERE parameters common to all sensitivity experiments is provided
in Table 2. For a full description of these parameters, the interested reader is referred to the CHIMERE user manual available
at http://www.lmd.polytechnique.fr/chimere.





Along the lateral boundaries of the coarse domain, the concentrations of the chemical species required by CHIMERE are provided by three-hourly forecasts of the ECMWF Composition Integrated Forecasting System (C-IFS) initialized at 00 UTC (Flemming et al., 2015). This global model comprises 60 vertical levels and has a horizontal resolution of $\approx 80 km$. In case a chemical species required by CHIMERE is not provided by C-IFS, the monthly climatological mean values from the MACC reanalysis (Inness et al., 2013) are used instead. As an exception, sea-salt aersols from MACC are applied albeit they are

also available from C-IFS because the latter system was found to overestimate the corresponding concentrations in our study region. This bias is of minor importance for the summer season considered here, but would lead to an overestimation of the PM concentrations in the other, stormier seasons of the year. Similarly, the applied dust aersols from C-IFS are scaled by a factor of 0.6 in order to compensate the positive bias observed during the two Saharan dust events occurring in the time period considered here. For all other chemical species from C-IFS, a scaling factor of 1 (i.e. no scaling) is used. The fact that the

chemical and physical boundary conditions for our CHIMERE forecasts come from different prediction systems is assumed to be of minor importance for the short leadtimes analysed here (27 hours from initialization at the utmost).

  To eliminate unwanted effects related to the spin-up, the daily WRF forecasts are initialized with the Digital Filtering Initialization (DFI) technique (Skamarock et al., 2008) and the first 3 integration hours are *not* used as meteorological input to CHIMERE. Consequently, CHIMERE is initialized on 03 UTC, using initial conditions from the model execution of the

previous day, and is then integrated until 03 UTC of the following day to complete one forecast day. This procedure is repeated for each day from June 20, 2018 to August 21, 2018 and the resulting model output is then concatenated to form time series covering the entire time period. Verification against surface observations as described in Section 2.5 begins on June 21st 03 UTC, so CHIMERE is permitted to spin-up during the first 24 hours of the integration.

## 2.2 Anthropogenic Emission Inventories, Landuse Databases and Postprocessing

To assess CHIMERE's combined sensitivity to changes in the anthropogenic emissions, downscaling strategy and landuse database, two distinct inventories and postprocessing techniques were selected: The EMEP dataset for the year 2017 on the one hand (EMEP/CEIP, 2019) and the HTAP v2.2 dataset for the year 2010 on the other (Janssens-Maenhout et al., 2015), both provided on a regular $0.1° \times 0.1°$ latitude-lonitude grid. To disaggregate the raw data from these inventories, the publicly available program emiSURF shipped with the CHIMERE source code was used (Mailler et al., 2017), which was here modified

to process EMEP data on the recently published $0.1° \times 0.1°$ grid. Spatial disaggregation is achieved by downscaling the emissions from their native grid to an auxiliary high-resolution grid at 1 km, followed by an upscaling to the two target domains displayed in Figure 1a. In the downscaling step, different proxies can be used to redistribute the raw emission data on the subgrid-scale, among which landuse categories are the standard option of the emiSURF program.

  To downscale the raw emissions from the HTAP v2.2 inventory, landused categories from the United States Geological

Survey (USGS, Loveland et al. (2000)) were used as the only proxy except for the "population downscaling" experiment, for which population density was used as an additional proxy (Gallego, 2010). Note that this kind of downscaling affects the $NO_2$ and particulate matter emissions from SNAP sector 2, originating mainly from domestic fuel burning.





To spatially regrid the EMEP inventory, road traffic density and the locations of large point sources were used *in addition* to population density and landuse categories, the latter provided by the GlobCover dataset (Bicheron et al., 2011). The road traffic
proxy affects the magnitude and allocation of the $NO_2$ emissions caused by this kind of activity whereas the locations of large point sources were used to re-allocate the corresponding emissions on the subgrid scale. The temporal disaggregation of the raw anthropogenic emission data to the timescale required by CHIMERE was accomplished by the use of seasonal, weekly and hourly profiles for each pollutant and activity sector, based on the standard information in the CHIMERE pre-processors (Mailler et al., 2017).

The above explained large differences between the spatial downscaling procedures of the two emission inventories were applied intentionally to assess CHIMERE's performance for the use of an up-to-date and sophistically downscaled inventory (EMEP) versus an older inventory downscaled with basic parameters (HTAP v2.2). For ease of understanding, these will hereafter be referred to as "emission configuration 1" and "emission configuration 2" respectively.

### 2.3   Specific Configuration of the Sensitivity Tests

To explore the influence of *vertical* resolution on model performance, 10 layer experiments are compared to 20 layer experiments, the lowermost layer being located at 999 hPa and the uppermost at 500 hPa in all cases (see Figure 1c+d). Thus, an increase in vertical resolution refers to a refinement in the lower to middle troposphere. An extension of the model top to, e.g., 200 hPa has been proposed in previous studies since some dust intrusions may extend to pressure levels above 500 hPa (Bessagnet et al., 2017). However, by design of our experiments, most of the dust intrusions' trajectory is simulated by the
global atmospheric composition model providing the lateral boundary conditions (C-IFS) rather than internally simulated by CHIMERE and, therefore, elevating the model top is assumed to be of minor importance here.

The effect of an increase in *horizontal* resolution is tested by comparing the model output obtained with the coarse resolution domain with that of the fine resolution domain nested therein (see Figure 1a,e and f). In all but one fine resolution experiment (the "coarse meteorology" experiment defined below) the horizontal resolution increase is undertaken in *both* CHIMERE and
WRF, meaning that the combined effect is assessed. Finally, the 2 horizontal and 2 vertical configurations are run seperately with emissions configuration 1 and 2 as defined in Section 2.2.

Version 2017r4 of the CHIMERE model offers the possibilty to use three distinct "chemical mechanisms" describing the gas-phase chemistry considered by CHIMERE. The "full Melchior" mechanism consists of 300 reactions and 80 gaseous species and is the most complete but also most computationally demanding of three. This is why a reduced version with
120 reactions and 40 species, the so called "reduced Melchior" or "Melchior 2" mechanism, is available as well. From version 2016a onwards, the SAPRC mechanism Carter (2010) is implemented as the third mechanism, offering a chlorine chemistry not considered in any of the two Melchior mechanisms (Mailler et al., 2017). With 72 gaseous species and 218 chemical reactions, SAPRC's complexity and computational costs are somewhat lower than for full Melchior, but far superior to reduced Melchior. For the European summer 2015, *reduced* Melchior and SARPC have been compared in Menut et al. (2013b), who found large
differences in the composition of organic nitrogen between the two which could potentially influence the spatial distribution of ozone production. They also found that the systematic overestimation of surface ozone reported in mainy CHIMERE studies





is slightly less a problem when using SAPRC. In the present study, however, the *full* version of the Melchior mechanism is applied instead of the reduced one, meaning that the aforementioned findings might not hold here.

Finally, three additional sensitivity tests are applied with constant anthropogenic emissions (HTAP), horizontal and vertical
resolution (fine mesh, 20 layers), chemistry mechanism (full Melchior) and landuse database (USGS). First, the effects of using the population proxy for downscaling the raw HTAP emissions are explored in what is called the "Population Downscaling" experiment (FM20H-P) hereafter. Then, the fine horizontal CHIMERE mesh is run with the *coarse* WRF mesh in the "Coarse Meteorology" experiment (FM20H-C) in order to see whether low resolution meteorological input deteriortates CHIMERE's performance. Finally, the effects of missing biogenic emissions are explored by intentionally turning them off in the "No
Biogenic Emissions" experiment (FM20H-N).

An overview of all applied sensitivity tests is provided in Table 3. In the last column, the computational costs for a typical summertime heat day (August 5th, 2018) are listed for the emission configuration 1 experiments. The runtimes of the respective configuration 2 experiments are in very close agreement (e.g. for CS10H and CS10E) but cannot be exactly stated since they were unfortunately not saved.

**2.4 The Air Quality Monitoring Network in Northwestern Spain (Galicia)**

The Galician air quality monitoring network comprises a total of 46 stations which, as a function of the main pollution source or the lack thereof, can be grouped into background, industrial and traffic sites (see Figure 1b). Currently, 14 stations are directly maintained by the Galician regional government (Xunta de Galicia). The remaining 32 stations are maintained by industrial companies which are supervised by the government in order to assure the same measurement standards, specified in
the national UNE-EN norm.

The quality control of the corresponding data is accomplished *manually* by trained technical staff of the regional government, i.e. is centralised in one institution. First, outlier values are detected by comparing a suspicious value to the typical time series behaviour at the considered site and at the surrounding sites. Once the outlier is detected, its validity is determined taking into account inter-variable relationships, potential power breakdowns, calibration errors, damages and changes in the topographic
features surrounding the station. This way, a quality controlled observational dataset has been developed which, at some locations, is now nearly a decade long. This dataset serves as reference for model verification.

**2.5 Applied Verification Measures**

Here, the *temporal* agreement between the modelled and observed time series is measured in terms of the Pearson correlation coefficient (R), the percentage bias (see Equation 1), and the standard deviation ratio (see Equation 2):

$$BIAS = \frac{\overline{m} - \overline{o}}{\overline{o}} \times 100 \qquad (1)$$





$$RATIO = \frac{\sigma_m}{\sigma_o} \tag{2}$$

, where $\overline{m}$, $\overline{o}$, $\sigma_m$ and $\sigma_o$ are the modelled and observed values for the temporal mean and standard deviation, respectively.

These measures are applied separately for the daily maximum, minimum and hourly time series of $NO_2$, $O_3$, $PM10$ and $PM2.5$. Note that the chosen verification measures are complementary to each other since they cover different time series aspects. Namely, BIAS and RATIO measure the model's capacity to reproduce the observed temporal mean and dispersion whereas R looks at the similarity in day-to-day variability irrespective of errors in the mean and dispersion. The perfect scores for BIAS, RATIO and R are 0, 1 and 1, respectively.

In addition, the mean absolute error (MAE) is a good measure of overall performance, and is here applied as a skill score (mean absolute error skill score, MAESS), i.e. as percentage deviation from the error of a reference experiment:

$$MAESS = \left(1 - \frac{MAE_i}{MAE_{ref}}\right) \times 100 \tag{3}$$

, where $MAE_i$ is the error a specific experiment $i$ and $MAE_{ref}$ the error of the experiment CS10E, used as reference throughout the present study since it is the computationally least expensive experiment (see Table 3). Positive values indicate performance gains, negative values performances losses with respect to the reference (Jolliffe and Stephenson, 2012). These verification measures are applied to hourly mean observations and hourly model data as provided by CHIMERE and, also, to the daily minimum and maximum values obtained from the former. All verification results are for the lowermost model layer whose upper limit is located at 999 hPa, i.e. roughly 10m above ground.

The aforementioned temporal verification scores are calculated separately for each station exceeding the 80% threshold of valid values and are then visulized either by overlay maps or boxplots. The centre line of each boxplot refers to the median value of the group of point-wise temporal verification results and the box to the interquartile range (IQR) of this group. The whiskers extend from the 25th percentile minus $1.5 \times$ IQR at the lower end to the 75th percentile plus $1.5 \times$ IQR at the upper end. Outlier verification results lying beyond these limits are not shown since their inclusion would blow up the scale of the figures and thus hamper their interpretabilty.

Apart from these temporal verification scores, the spatial bias (SBIAS, S = "spatial"), correlation coefficient (SR), standard deviation ratio (SRATIO) and mean absolute error (SMAE) were calculated on the pointwise temporal *mean* values in order to assess whether the spatial statistics of the *average* pollutant concentrations are captured by the model. Likewise, the same scores have been applied on the pointwise temporal *standard deviation* values to assess whether the model reproduces the spatial statistics of *temporal variablity*.





## 3  Results

### 3.1  Maximum Values

**3.1.1  Temporal Mean and Standard Deviation**

Fig. 2 shows the temporal *mean* values of the daily *maximum* concentrations seen in observations (the dots) plotted on the respective model value (the underlying pattern) for the 4 experiments driven with emission configuration 1 and the chemical mechanism SAPRC (CS10E, CS20E, FS10E and FS20E, row 1-4). Rows are ordered so that the first pair refers to the coarse horizontal mesh and the second to the fine one. Further, the 10 and 20 vertical layer experiments are placed on top of each
other to assess the effects of an increase in vertical resolution. In the fifth row, the fourth experiment (high hores, 20 layers) is replicated with emission configuration 2 to show the effects of a combined change in the choice of the anthroponenic emission inventory (from EMEP to HTAP), downscaling technique (from landuse, populatuion and traffic downscaling to landuse downscaling only) and landuse database (from Globcvoer to USGS). In the header of each subplot, the spatial bias (SBIAS, in ug/m3), correlation coefficient (SR), standard deviation ratio (SRATIO = $\sigma_{model}/\sigma_{obs}$) and mean absolute error
(SMAE) of the modelled vs. observed temporal mean values is provided.

As can bee seen from Figure 2, an increase in *horizontal* resolution improves the model's performance for $PM2.5$ by reducing SBIAS and by bringing SRATIO closer to unity. For $NO_2$ and $O_3$, however, model performance either does not improve or clearly deteriorates. Most notably, SBIAS and SRATIO increase, the latter exceeding a value of 2, which means that the spatial dispersion of the modelled mean $NO_2$ values is more than twice the observed one. As will be shown below (see
Section 3.1.2), these error increases are likely associated with the population downscaling technique used to disaggregate the raw EMEP emissions.

An increase in *vertical* resolution reduces SBIAS by up to 2.4 ug/m3 (i.e. 40%) for the mean $O_3$ values and by up to 1.0 ug/m3 (i.e. 83%) for the mean $PM2.5$ values. For the latter pollutant, vertical refinement is much more efficient when using the fine horizontal mesh. However, these improvements are limited to SBIAS and do not affect the other spatial performance
measures.

For the fine horizontal mesh and 20 vertical layers, a switch to emission configuration 2 (i.e. from FS20E to FS20H, compare rows 4 and 5) translates into an improvement of SRATIO for $NO_2$ and $O_3$ but to a worsening for $PM2.5$. Also, results for FS20H are in closer agreement with CS20E than with FS20E, which points to the fact that the temporal *mean* values are more senstitive to the particular setup of the downscaling technique than to the sole differences in raw inventories.
In all considered experiments, the simulated mean $O_3$ concentrations are considerably higher over the sea than over land, which is in line with Terrenoire et al. (2015) and can be explained by reduced dry deposition and nighttime destruction by $NO_2$ over the sea resulting from a reduced surface roughness and $NO_2$ availability there (Davies et al., 1992; O'Hare and Wilby, 1995). Since this land-sea contrast is not seen in observations, the SR values for all experiments is essentially zero. This can be either explained by the lack of off-shore background observations (note that all available coastal sides are affected by urban





pollution) or by the fact that the reduced ozone destruction over the sea is less pronounced in the model than in the real world, translating into a positive bias there.

Figure 4 shows the temporal *standard deviation* of the daily *maximum* concentrations as seen in observations vs. those seen in the model, i.e. the model's capability to reproduce the observed temporal variability. In general, CHIMERE tends to underestimate this kind of variability, i.e. is plagued by underdispersion (SBIAS is negative). An increase in *horizontal*

resolution alleviates this problem for $PM2.5$ and even leads to a pronounced overdispersion for $NO_2$ (i.e. to a positive SBIAS) but does not noticeably alter the results for $O_3$. For $PM2.5$, SR is much improved when considering the fine horizontal mesh. Contrary to the findings for the temporal mean, temporal variablity is more sensitive to a horizontal resolution increase than to a verical resultion increase. Except for P$PM2.5$, the impact of a switch in the emission configuration is less pronounced for the temporal standard devation than for the aformentioned temporal mean (compare rows 4 and 5 in Fig. 3 and 4)

**3.1.2   Full Temporal Verfication**

Fig. 5 shows the verficiation results of *all* applied experiments as ordered in Table 3) for the daily *maximum* $NO_2$ and $O_3$ concentrations. The perfect score for a given verification measure is indicated by a red vertical line. As can be seen from the figure, the $NO_2$ concentrations are generally underestimated by the model, except for the four emission configuration 1 experiments run on a high horizontal resolution (see Fig. 4a). Emission configuration 2 is plagued by larger median biases

(see vertical orange lines within the boxes) than configuration 1 but has the advantage of a lower spatial spread in the results (see width of the boxes and whiskers). When applying a high horizontal resultion, this bias is reduced on average (see median values) but the aforemantioned spread is largely increased. While the effects of a vertical resolution increase and/or switch in the applied chemical mechanism are negligible, the effect of population dowscaling is considerable. Namely, the smallest median bias and largest spatial spread among all experiments is yielded if the raw HTAP emissions are disaggregated this way

(see FM20H-P in Fig. 4a).

The structure of the verfication results for the standard deviation ratio (see Equation 2) is in very close agreement with the aforementioned structure found for the percentage bias and virtually indentical lessons are learned (see Fig. 3a+b).

The model's capablity to simulate the temporal sequence of the observations, here measured with the Pearson correlation coefficient (R), is most improved by an increase in the horizontal resolution (see Fig. 4c). Emission configuration 1 yields

systematically better results than configuration 2 (compare experiments ending on E with those ending on H in Fig. 4c). As opposed to the bias, the spatial spread of the *correlation coefficient* is larger for the coarse horizontal resolution than for the fine one, particularly if emission configuration 1 is used (compare the spread of the "C..." type experiments in Fig. 4a+c). The full Melchior mechanism yields slightly better correlation coefficientsis than SAPRC and so does the use of 20 instead of 20 vertical layers (see 4c).

As indicated by Fig. 4d, the MAESS of the reference experiment CS10E is improved only by the CM20E experiment, meaning that the use of 20 vertical layers together with the full Melchior mechanism is sufficient to achieve optimal results for this measure. A horizontal resolution increase is not necessary and is actually counterproductive if emission configuration 1 is used.





The inclusion of the population proxy in the downscaling procedure of the HTAP inventory leads to a sharp decrease in the
spatial median MAESS and to the largest spatial spread among all experiments (see FM20H-P in Fig. 4d). In comparsion, the
use of coarse meterological input data or removal of biogenic emissions has much smaller effects on the model's performance
(compare FM20H-C and FM20H-N with FM20H in Fig. 4d)

As shown in Fig. 4e and f, CHIMERE overestimates the temporal mean and underestimates the temporal variability of the
daily maximum $O_3$ concentrations. The effect of the performance factors is similar for each of the four applied verification
measures. In general, the respective error is improved by a vertical resolution increase and by applying full Melchior instead
of SAPRC, but is deteriorated or not improved when the horizontal resolution is increased. As an exception, SAPRC generally
yields better correlation coefficients if the fine horizontal mesh is used (see 4g). Contrary to Menut et al. (2013b), average
$O_3$ concentrations are larger for the SAPRC mechanism than for full Melchior. When considering MAESS, the emission
configuration is the most influential factor on model performance, with configuration 1 clearly outperforming configuration
2 (see Fig. 4h). As was the case for maximum $NO_2$, 20 vertical layers yield better results than 10 layers and, for the use
of emission configuration 1, the 20 layer setup performs nearly as well for the coarse horizontal mesh than for the fine one,
meaning that the former is again preferable in case computational resources are limited (see last column in Table 3).

The full temporal verification results for the daily maximum $PM2.5$ and $PM10$ concentrations are displayed in Fig. 5.
As shown in panels a+b and e+f, CHIMERE generally understimates the temporal mean value and the temporal variability
for both size fractions. The most important peformance factor is the emission configuration, yielding smaller bias values with
configuration 1 (see Fig. 5a+e) and better correlation coefficients with configuration 2, particularly for the fine particles (Fig.
5c+g). The effects of a horizontal resolution increase depend on the considered emission configuration and particle size fraction.
Namely, configuration 1 improves the bias and standard deviation for both size fractions (Fig. 5a+b and e+f) but has no effect
on the correlation coefficient (Fig. 5c+g). Configuration 2, in turn, improves the correlation coefficient of the fine particles
(Fig. 5c), but does not affect the bias nor the standard deviation ratio for any of the two particle size fractions (5a+b and e+f).
A vertical resolution increase improves the bias for both particles sizes and, if a fine horizontal mesh is applied in addition,
also the standard deviation ratio for the fine particles. The correlation coefficient, however, cannot be improved by this kind
of resolution increase and even deteriorates for some experiments (Fig. 5c+g). Regarding overall performance as measured by
the MAESS (5d+h), SAPRC yields better results than full Melchior for nearly all experiments and both size fractions. The
most robust skill increases are *again* obtained with 20 vertical layers, the coarse horirzontal resolution, the SAPRC mechanism
and emission configuration 1 (CS20E). Albeit the performance increase at individual stations may be much larger for other
experiments, CS20E yields positive MAESS values at *all* stations and for *both* particles sizes. If the fine horizontal resolution
is used (FS20E), the average performance improves for $PM10$ but deteriorates for $PM2.5$. FS20H and FM20H-P perform
equally well than CS20E on average, but are characterized by a larger spatial spread in the results.

The population downscaling experiment outperforms its base experiment or is comparable to it for both particle sizes (com-
pare FM20H-P with FM20H in all panels of Fig. 5). Using coarse resolution meteorlogical input does not noticeably affect the
results, except for a clear decrease in correlation for the fine particles (compare FM20H-C with FM20H in Fig. 5c). A lack of





biogenic emission, however, largely enhances the bias (compare FM20H-N with FM20H in Fig. 5a+e), reduces the correlation (Fig. 5c+g) and worsens the overall performance as measured by the MAESS (Fig. 5d+h).

## 3.2    Minimum Values

### 3.2.1    Temporal Mean and Standard Deviation

Fig. 6 shows the temporal *mean* values of the daily *minimum* concentrations seen in observations (the dots) plotted on the respective model value (the underlying pattern) for the five experiments assessed in Section 3.1.1. For $NO_2$ (Fig. 6, column 1), the model underestimates the temporal mean concentrations on average (SBIAS < 0) and underestimates their spatial dispersion
(SRATIO < 1). The spatial pattern of the observed mean values is also not well reproduced by the model (SR < 0.25 in Fig. 6a, d, g, and j). While the former two error types can be improved by augmenting the horizontal resolution (compare panels a+d with panels g+j in Fig. 6), the latter one can be reduced by using emission configuration 2 (compare panel j with m). Similar to the results for the maxima, using 20 instead of 10 vertical layers does not noticeably improve the result for the $NO_2$ *minima* either (compare Fig. 6a with d and g with j).

As for the maxima, the average *minimum* $O_3$ concentrations in (Fig. 6, column 2) are overestimated by the model (SBIAS > 0 in column 1). However, the spatial pattern of the observed values is generally well reproduced (RS ≥ 0.65) and so is the spatial dispersion if the coarse horizontal mesh is used (SRATIO ≈ 1). Using the fine horizontal mesh on the one hand reduces the bias but, on the other, inflates the spatial dispersion (SRATIO > 1, compare Fig. 6b with h and e with k). Results are improved when 20 instead of 10 vertical layers are used with the horizontal mesh (compare panels h and k) and deteoriorate
when emission configuration 1 is applied (compare panels k and n).

The temporal mean $PM2.5$ values (Fig. 6, column 3) are on average overestimated by the model (SBIAS > 0), their spatial dispersion is underestimated (SRATIO well below unity) and their spatial pattern not well reproduced (low values for SR). A *horizontal* resolution increase improves the spatial dispersion but deteriorates the spatial pattern and increases the bias, meaning that the negative effects prevail for this factor (compare Fig. 6 c with i and f with l). A *vertical* resolution increase
generally has little effects on the model's performance unless the horizontal resolution is as well increased, in which case the bias worsens for $PM2.5$ (compare c with with f and i with l). As for the maxima, results for FS20H are generally more similar to CS20E than to FS20E.

Fig. 7 shows the respective verfication results for the *temporal standard deviation* of the daily minimum concentrations. For $NO_2$ (first column), the model on average underestimates the temporal variability (RBIAS < 0) and the associated spatial
dispersion (SRATIO well below unity). With SR values ranging in between 0.35 and 0.56, some skill is obtained for the spatial distribution of temporal variability. Results are insensitive to a *vertical* resolution increase (compare Fig. 7a with d and g with j) but systematically improve if the *horizontal* resolution is augmented (compare a with g and d with j). The temporal variabilty of the $O_3$ minima (Fig. 7, column 2) is on average well reproduced by the model (SBIAS ≈ 0). However, the associated spatial distribution is missed (SR ≈ 0) and the dispersion overestimated (SRATIO > 1). Neither a horizontal nor a vertical resolution
increase nocticeably improves these results. The temporal variability of the $PM2.5$ minima (Fig. 7, column 3) is also well





reproduced on average and some skill is obtained for the respective spatial distribution. As for the $NO_2$ minima, the degree of spatial dispersion is as well underestimated for the $PM2.5$ minima and can be improved by a horizontal resolution increase (compare panels c with i and f with l). Results for FS20H closely agree with those for FS20E, except for generally lower $O_3$ and higher $PM2.5$ concentrations (compare the last two rows in Fig. 7).

### 3.2.2 Full Temporal Verfication

Fig. 8 shows the full temporal verfication results for the daily *minimum $NO_2$* and $O_3$ concentrations. For a correct interpretation of the results, it is here important to note that the observed minimum concentrations in our study region are generally low and that average differences of only a few ug/m3 can translate into large *percentage* bias values.

As can be seen from Fig. 8a+b, the temporal mean and standard deviation of the daily minimum $NO_2$ concentrations are considerably underestimated at nearly all stations in any of the applied experiments. The spatial median values for BIAS and RATIO can be improved with a horizontal resolution increase and either emission configuration 1 (FS10E, FM10E, FS20E and FM20E) or configuration 2 plus population downscaling (FM20H-P), implying that this kind of downscaling is key at this point. However, improvements in the spatial median can only be achieved at the expense of a large increase in the spatial spread of the results, which is in line with the findings obtained for the $NO_2$ maxima (see Section 3.1.2). For the correlation coefficient (Fig. 8c), emission configuration 1 performs better than configuration 2, full Melchior better SAPRC and the coarse horizontal mesh better than the fine one. In comparision, an increase in vertical resolution from 10 to 20 layers is less efficient in improving the correlation. Coarse resolution meterological input data and missing biogenic emissions both slightly worsen the model performace for all applied verification measures (compare FM20H-C and FM20H-N with FM20H in panels a, c, e and g). When considering the MAESS (Fig. 8d), the spatial median performance for the base experiment (CS10E) cannot be improved by any of the applied alternative experiments and the aforementioned growth in the results' spatial dispersion due to population downscaling can be clearly seen for FM20H-P.

Similar to the respective results for the maximum concentrations, daily *minimum $O_3$* concentrations are also on average overestimated by the model (Fig. 8e) and the results for all verification measures can be improved by applying the full Melchior mechanism and 20 vertical layers (Fig. 8e to h). Contrary to the maxima, the spatial median performance for the $O_3$ minima can be generally further improved by applying a fine horizontal mesh, the downside of an increased spatial spread being less pronounced than for the maxima. The overall performance in terms of MAESS (Fig. 8h) is very satisfactory for the coarse horizontal resolution experiments run with 20 vertical layers (see CS20E and CM20E), which is in line with the results for the maxima. However, due the relatively low spread increase mentioned above, running the fine horizontal mesh —perferably with emission configuration 1— is more tentative for the $O_3$ minima than for the maxima (compare CS20E, CM20E with FS20E and FM20E in Fig. 8h and 4h). Coarse resolution meterological input data or missing biogenic emissions both have negligible effects on the results. Population downscaling, however, leads to a systematic improvement (compare FM20H-C, FM20H-N and FM20H-P with FM20H in Fig. 8h).

The full temporal verifation results for the $PM2.5$ and $PM10$ minima are displayed in Fig. 9. The model systematically overestimates the temporal mean $PM2.5$ concentrations and also tends to overestimate the temporal variability (Fig. 9a+b).



Using 20 vertical layers instead of 10 enhances the correlation coefficient on the one hand but on the other generally increases the bias and shifts the standard deviation ratio to values larger than unity (except for moving from CS10E to CS20E, see Fig. 9a,b,c). A horizontal resolution increase has similar effects which are, however, larger in magnitude. Switching from SAPRC to full Melchior improves the results for all measures and nearly all experiments and overall performance gains as measured by MAESS are largest for this kind of switch (see panels a to d). When spatial median values are considered, the

MAESS obtained with emission configuration 2 are systematically better than those obtained with configuration 1 (see Fig. 9d). However, the spatial spread in the MAESS is larger for configuration 2 than for configuration 1. In comparision with FM20H, overall performance deteriorates for the population downscaling experiment (see FM20H-P) and, even more so, for the coarse meteorological input experiment (see FM20H-C). Missing biogenic emissions improve the MAESS on average, but also increase the spatial spread (see FM20H-N). Notably, the performance increase of the CM10E experiment (with respect

to the base experiment CS10E) is positive at *every* station, which is rarely the case in the present study. Hence, the coarse *horizontal* mesh is again a straightforward option which already yields optimal results with a simple 10-layer setup for the simultaneous use of the full Melchior mechanism.

For the $PM10$ *minima*, emission configuration 2 yields smaller bias values and more favourable standard deviation ratios than configuration 1 (Fig. 9e+f), but weaker correlation coefficients (panel g). Using full Melchior instead of SAPRC and 20 instead

of 10 vertical layers reduces the bias for all experiments, both factors being of roughly equal importance for this pollutant and temporal aggregation. Correlation coefficients are also improved, but only for the experiments run with emission configuration 1. If emission configuration 2 is used, SAPRC yields roughly the same correlation coefficients than full Melchior (Fig. 9g). The standard deviation ratios are systematically better for SAPRC than for full Melchior and for 20 instead of 10 layers if the fine horizontal mesh is chosen. Regarding MAESS (Fig. 9h), performance losses caused by population downscaling or

coarse resolution meterological input are less pronounced for the coarse particles than for the fine ones (compare FM20H-P and FM20H-C with FM20H in Fig. 9d+h). As for the fine particles, the "no biogenic emissions" experiment is also plagued by an increased spatial variablity in the MAESS and, unlike the results for the fine particles, suffers a spatial average performance *decrease* if compared to its base experiment (compare boxes and median values for FM20H-N with FM20H in Fig. 9d+h). As expected, the modelled mean values are more realistic when biogenic emissions are taken into account (compare FM20H with

FM20H-N in Fig. 9e). As for the fine particles, optimal results are obtained with the coarse horizontal mesh run with only 10 layers and the full Melchior mechanism (see CM10E in panel Fig. 9h). Albeit it is second choice for the fine particles, emission configuration 1 is preferable to configuration 2 for the coarse particles.

### 3.3   Verification Results per Pollution Source

Figure 10 shows the spatial median MAESS with reference to the base experiment CS10E for all locations (row 1) and sep-

arately for background, industry and traffic locations (rows 2 to 4). The first column refers to the results for daily maximum concentrations, the second to hourly concentrations and the third to daily minimum concentrations, respectively. Improvement over the base experiment is indicated by green, worsening by red colour shadings.





As can bee seen from the predominantly red color shadings in the first two columns of Fig. 10, the base experiment CS10E already provides a good overall skill, difficult to exceed when considering daily maximum or hourly concentrations. Among all 440 suggested model improvement factors, the use of 20 instead of 10 vertical layers yields the most balanced increases in spatial *median* performance irrespective of the applied chemical mechanism (see CS10E and CM20 in these columns). Switching from the coarse to high *horizontal* resolution leads to large performance increases for *particular* pollutants and/or station types, but only at the expense of performance decreases for the remaining species and sides and thus to unbalanced results.

Irrespective of the applied emission configuration and number of vertical layers, the best results for the *maximum and hourly* 445 $NO_2$ *values* are obtained with a *coarse horziontal resolution*, except at traffic stations where the fine horizontal mesh yields better results if, importantly, emission configuration 2 is used *without* population downscaling (compare FS10E, FM10E, FS20E and FM20E in Figure 10j and k). At traffic and industry sides, the worst results for the $NO_2$ maxima and hourly data are obtained with the fine horizontal mesh and emission configuration 1 (relying on population and traffic downscaling) and with configuration 2 plus population downscaling (note the similarity between FS10E, FM10E, FS20E, FM20E and FM20H-P 450 in Fig. 10a,b,g,h,j,k). Hence, this kind of downscaling is not advantegeous in these cases.

For daily minimum $NO_2$, the coarse horizontal resolution is again the best chocice, but only in combination with emission configuration 1 (see CS10E, CM10E, CS20E and CM20E in panels c, f, i and l). Using the coarse horizontal resolution with configuration 2 instead yields heterogenous results, i.e. a large performance increase at industrial sides contrasted by a large decrease at traffic sides (compare CS10H, CM10H, CS20H and CM20H in panel i with panel l).

For $O_3$, emission configuration 1 performs systematically better than configuration 2. Among the emission configuration 2 experiments, it is again the populuation downscaling experiment that most closely resembles the results from the configuration 1 experiments (compare experiments ending on "E" with FM20H-P). Importantly, using 20 instead of 10 vertical layers yields performance gains in virtually *any* case, i.e. irrespective of the applied emission configuration, horizontal mesh, chemical mechanims, temporal aggregation and pollution source type, and is consequently the most *robust* model improvement factor 460 for surface $O_3$ concentrations assessed here. Second best in this context is the use of the full Melchior mechanism instead of SAPRC. Note also that the results for the maxima and hourly data are more dissimilar to each other than for the remaining pollutants.

As opposed to the findings for $O_3$, emission configuration 2 is the better choice for $PM2.5$, particulary considering daily minimum concentrations at all kind of sides, as well as as maximum and hourly concentrations at industrial and traffic sides. 465 The effects of a vertical resolution increase are heterogeneous. At background sides (see second row in Fig. 10 and also Supplementary Figure 2), results are improved for the daily maxima but deteriorate for the minima, with very little effects on the results for hourly concentrations. At industrial and traffic sides, however, results generally worsen for this factor. At background sides, SAPRC is generally superior to full Melchior whereas the opposite is found at industrial and traffic sides. As for $O_3$, the horizontal resolution increase is not advantageous for $PM2.5$ either, except for the daily minimum concentrations 470 at industrial and traffic sides when using emission configuration 1.

The applied factor changes are generally less effective for $PM10$ than for the other three pollutants. Largest performance gains are obtained for daily maximum concentrations, particuluary at traffic sides, if the fine horizontal mesh is used in





combination with 20 vertical layers and emission configuration 1 (see experiments FS20E and FM20E in panels a, d, g, and
j). The same mesh, however, yields largest performance *losses* for minimum concentrations at background sides if emission
configuration 2 is applied (see panel f). Albeit the differences are generally weak, the SAPRC mechanism is preferable for
maximum and hourly concentrations whereas full Melchior is preferable for the minima.

Among the three specific sensitivity experiments, the "population downscaling" (FM20H-P) experiment exhibits the largest
performance deviations from their common base experiment (FM20H), followed by the "no biogenic emissions" (FM20H-
N) and "coarse meteorology" (FM20H-C) experiments. FM20H-P performs particularly bad for maximum and hourly $NO_2$
concentrations at industy and traffic sides (see panels g, h, j and k) and particularly well for minimum $O_3$ concentrations at
traffic sides (see panel l). Curiously, among all considered experiments, FM20H-N yields the best results for minimum $PM2.5$
concentrations at industry and traffic sides (see panels i and l) and for maximum $NO_2$ concentrations at traffic sides (see panel
j). The good skill scores at these stations types arise from error compensation effects. Namely, the positive bias is for minimum
$PM2.5$, which is smaller at traffic and industry sides than at background sides because the observed concentrations are higher
there, is *improved* when biogenic emissions are turned off, which translates into better MAESS values. For maximum $NO_2$,
removing this kind of emissions enhances the temporal correlation, brings the standard deviation closer to unity and finally also
improves the MAESS. This, in turn, means that the *inclusion* of biogenic emissions in the remaining experiments deteriorates
the temporal variability and day-to-day sequence of the modelled minimum $NO_2$ time series if compared with observations.
At background sides, however, the $NO_2$ and $PM2.5$ maxima are generally underestimated by the model and the exclusion of
biogenic emissions further increases this negative bias (see Fig. 10d and Supplementary Figure 2). The pronounced reduction
of the $O_3$ maxima at background sides in the FM20H-N experiment, as compared with FM20H, points to an active role of
biogenic VOCs in this case (see Supplementary Figure 2e). For FM20H-C, deviations from the base experiment are largest for
the minima at industry sides and are otherwise generally weak (see Fig. 10i).

## 4 Discussion and Conclusions

In this study, a series of 19 sensitivity experiments was carried out with the chemical weather forecasting model CHIMERE
over the northwestern Iberian Peninsula for the 2018 summer season in order to assess the model's capablitiy to reproduce in-
situ $NO_2$, $O_3$, $PM10$ and $PM2.5$ surface concentrations on daily to hourly timescale. The range of applied model experiments
covers the effects of distinct emission configurations, horizontal and vertical resolution setups and model chemistries. With the
help of three secondary experiments, the impacts of population downscaling, coarse resolution meteorological input data and
missing biogenic emissions are discussed in addition. All these experiments were driven by meteorological data from WRF
and chemical boundary data from ECMWF C-IFS.

The obtained results are very heterogeneous and the applied model improvement efforts, often associated with consider-
able computational costs, do generally *not* lead to an *unrestricted* model improvement. For most efforts, verfication results
improve for *some* aspects but worsen for others. Nonetheless, one single factor has been identified that improves the model in
a systematic way, returning better results for virtually all aspects of the verification.





The first take-home message is that the use of an up-to-date and sophistically downscaled anthropogenic emission inventory (configuration 1: EMEP for the year 2017 downscaled with landuse, population and traffic proxies as well as large point sources), if compared to an older inventory downscaled with basic options (configuration 2: HTAP v2.2 for the year 2010 downscaled with landuse only), on the one hand improves the modelled $O_3$ and $PM10$ concentrations, but on the other hand

deteriorates the results for $NO_2$ and $PM10$. This is in line with Russo et al. (2019), in the sense that an upgraded emission inventory does not necessarily improve the modelled pollutant concentrations with respect to observations in all aspects.

Second, heterogeneous results are obtained for the performance changes associated with the *chemical mechanism*. While the performance for $NO_2$ is practically unrelated to the chosen mechanism, the full Melchior mechanim is pereferable to SAPRC if $O_3$ concentrations —at any temporal scale— are considered. For particulate matter, SAPRC is preferable for the daily maxima

and hourly concentrations and full Melchior for the daily minima.

Third, an increase in the *horizontal* resolution of the CHIMERE domain and associated emissions from $0.15° \times 0.15°$ to $0.05° \times 0.04°$ does *not* lead to a systematic model improvement but rather to a large increase in the spatial variability of the results. In line with Valari and Menut (2008), we have indications that this is caused by the noise increase in high resolution *meteorological* input data and, to an even larger degree, by the populuation downscaling procedure used to reallocate the raw

data from the applied anthropogenic emission inventories on the subgrid scale. If this kind of downscaling is used, the model overestimates the *temporal mean* value of the daily maximum and hourly concentrations at traffic and industry sides. The same applies to the *temporal standard deviation*, i.e. to the model's capabiltiy do simulate the degree of temporal variability from one day to another.

Contrary to the effects obtained with an increased horizontal resolution, the use of 20 instead of 10 vertical layers within the

lower to middle troposfere (999 to 500 hPa) *systematically improves* the model results for nearly all aspects of the verifcation.

All together, and as long as top-down emission inventories coming on a relatively coarse spatial and temporal resolution are applied, we recommend the use of 20 model layers together with a horizontal resolution not much finer than the native resolution of the inventory. In this context, the resolution of the coarse domain applied here ($0.15° \times 0.15°$) may not be optimal and in future studies should be approximated to the native grid of the emission inventory (i.e. $0.1° \times 0.1°$ for both HTAP and

EMEP) in order to see whether the results can be further improved. Likewise, a *region-specific* optimization of the downscaling procedures used to re-allocate raw emissions on the subgrid scale according to proxy data for population and traffic density would likely yield better results for the northwestern Iberian Peninsula, particularly for the $NO_2$ and $PM2.5$ concentrations.

As a final remark, the present study has explpored a *broad* range of model performance factors with *empirical* methods, mainly to provide practial recommendations for the numerical modelling community. In the future, our results should be

complemented by analytical in-depth studies focussing on single factors.

*Code availability.* The CHIMERE and WRF source codes are publicly available from http://www.lmd.polytechnique.fr/chimere and http://www2.mmm.ucar.edu/wrf/users/downloads.html, respectively.



*Data availability.* The CHIMERE and observational data underlying the verification results presented here have been made available by the authors and can be retrieved from the public Dropbox link https://www.dropbox.com/s/eb6y5pjsb6phbf2/Brands_et_al_2019_acpd_ underlying_data.zip?dl=0. The GFS data used to drive WRF and the C-IFS data used as lateral boundary conditions for the CHIMERE simulations can be downloaded from https://rda.ucar.edu/datasets/ds084.1 and ftp://dissemination.ecmwf.int, respectively. The HTAP v2.2 and EMEP 2017 emission inventories can be retrieved from https://edgar.jrc.ec.europa.eu/htap_v2/ and https://www.ceip.at/new_emep-grid/ 01_grid_data and their public availablity, as for remaining datasets used here, is kindly acknowledged.

*Author contributions.* Swen Brands designed and executed the CHIMERE experiments, disaggregated the HTAP emission inventory, built the figures, analysed the results and wrote the manuscript. Guillermo Fernández-García and Marcos Tesouro Montecelo provided the meteorological input from WRF and contributed to the manuscript. Marta García disaggregated the EMEP emission inventory and contributed to the manuscript, Nuria Fernández García, Anthony Estévez Saunders, Paula Costa Tomé and Cristina Otero were responsible for the quality control of the observations and contributed to the manuscript. Pablo Enrique Carracedo García, Anabela Neto Venancio, Pedro Daniel Melo Costa, María Luz Macho and Juan Taboada contributed to the manuscript.

*Competing interests.* The authors declare no competing interests.

*Acknowledgements.* The authors would like to thank the CHIMERE development team working at LMD Paris and also the WRF development team, for providing the models' source codes and technical support. A special thanks goes to Florian Couvedat and Bertrand Bessagnet for sharing their programs dedicated to the downscaling of raw anthropogenic emission with various types of proxy data. The authors also gratefully acknowledge the computational resources and technical support provided by the Centro de Supercomputación de Galicia (CESGA), as well as the free availability of the global predictions from the GFS and C-IFS forecasting systems maintained by NCEP and ECMWF/Copernicus, respectively.





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

**Figure 1.** (a) Horizontal CHIMERE domains used for all sensitivity experiments. The fine domain (orange rectangle) is nested into the coarser one (blue rectangle). At the lateral boundary conditions of the coarse domain, CHIMERE is fed by time varying C-IFS data. (b) The Galician air quality station network, grouped by the main pollution sources, (c) Height of the model layers in CHIMERE along 43 °N when using the 10 layer setup and the fine domain, (d) as c but for the 20 layer setup (e) model orography in metres above sea-level (a.s.l.) for the coarse domain, (f) as e but for the fine domain.



**Figure 2.** Observed (dots) vs. modelled (underlying pattern) temporal *mean* values for the daily maximum concentrations of $NO_2$, $O_3$ and $PM2.5$ and the 5 experiments marked with an asterisk in Table 3, all run with the SAPRC mechanism. Also shown is the spatial mean difference between the modelled and observed mean values (SBIAS), their correlation coefficient (SR), standard deviation ratio (SRATIO) and mean absolute error (SMAE).





**Figure 3.** Observed (dots) vs. modelled (underlying pattern) temporal *standard deviation* values for the daily maximum concentrations of $NO_2$, $O_3$ and $PM2.5$ and the 5 experiments marked with an asterisk in Table 3), all run with the SAPRC mechanism. Also shown is the spatial mean difference between the modelled and observed standard deviation values (SBIAS), their correlation coefficient (SR), standard deviation ratio (SRATIO) and mean absolute error (SMAE).





**Figure 4.** Temporal verification results for daily near-surface *maximum NO$_2$* (left) and *O$_3$* (right). Row 1: percentage bias (BIAS), row 2: Pearson correlation coefficient (R), row 3: ratio of standard deviations (RATIO), row 4: mean absolute error skill score (MAESS) with reference to the base experiment CS10E. Boxplots are calculated upon the point-wise verification results at all available stations. Experiments are explained and grouped as in Table 3.



**Figure 5.** Temporal verification results for daily near-surface *maximum PM*2.5 (left) and *PM*10 (right). Row 1: percentage bias (BIAS), row 2: Pearson correlation coefficient (R), row 3: ratio of standard deviations (RATIO), row 4: mean absolute error skill score (MAESS) with reference to the base experiment CS10E. Boxplots are calculated upon the point-wise verification results at all available stations. Experiments are explained and grouped as in Table 3.



**Figure 6.** Observed (dots) vs. modelled (underlying pattern) temporal *mean* values for the daily minimum concentrations of $NO_2$, $O_3$ and $PM2.5$ and for the five 5 experiments marked with in asterisk in Table 3), all run with the SAPRC mechanism. Also shown is the spatial mean difference between the modelled and observed mean values (SBIAS), their correlation coefficient (SR), standard deviation ratio (SRATIO) and mean absolute error (SMAE).





**Figure 7.** Observed (dots) vs. modelled (underlying pattern) temporal *standard deviation* values for the daily minimum concentrations of $NO_2$, $O_3$ and $PM2.5$ and the 5 experiments marked with an asterisk in Table 3, all run with the SAPRC mechanism. Also shown is the spatial mean difference between the modelled and observed standard deviation values (SBIAS), their correlation coefficient (SR), standard deviation ratio (SRATIO) and mean absolute error (SMAE).



**Figure 8.** Temporal verification results for daily near-surface *minimum NO₂* (left) and *O₃* (right). Row 1: percentage bias (BIAS), row 2: Pearson correlation coefficient (R), row 3: ratio of standard deviations (RATIO), row 4: mean absolute error skill score (MAESS) with reference to the base experiment CS10E. Boxplots are calculated upon the point-wise verification results at all available stations. Experiments are explained and grouped as in Table 3.

**Figure 9.** Temporal verification results for daily near-surface *minimum PM*2.5 (left) and *PM*10 (right). Row 1: percentage bias (BIAS), row 2: Pearson correlation coefficient (R), row 3: ratio of standard deviations (RATIO), row 4: mean absolute error skill score (MAESS) with reference to the base experiment CS10E. Boxplots are calculated upon the point-wise verification results at all available stations. Experiments are explained and grouped as in Table 3.





**Figure 10.** Spatial median mean absolute error skill score (MAESS) with respect to the base experiment CS10E for daily maximum, hourly or daily minimum concentrations (columns 1 to 3 respectively) at all available stations (row 1) or at background, industrial or traffic stations (row 2 to 4 respectively).





| Parameter | Option |
|---|---|
| Microphysics | WRF single-moment 6-class scheme |
| Longwave radiation | Rapid Radiative Transfer Model |
| Shortwave radiation | Dudhia scheme |
| Surface layer | MM5 similarity |
| Land surface | 5-layer thermal diffusion |
| Planetary boundary layer | Yonsei University scheme |
| Cumulus | Kain-Fritsch scheme |

**Table 1.** *WRF* physics common to all sensitivity tests





| Parameter | Option |
|---|---|
| Nr. Gauss-Seidel iterations | 3 |
| Chemical time-step | adaptive |
| Physical time-step | 5 minutes |
| Nr. of aerosol size sections | 9 |
| Chemically-active aerosols | yes |
| Sea-salt emission parameterization | inert, parametrization 0 |
| Biogenic emissions | MEGAN |
| Mineral dust emission | On |
| Saltation and sandblasting scheme | Alfaro and Gomes (2001), Menut et al. (2005) |
| Wind threshold estimation | Shao and Lu (2000) |
| Effect of soil moisture on mineral dust emissions | Fécan et al. (1998) |
| Secondary organic aerosol scheme | medium complexity |
| ISORROPIA coupling | yes |
| Inclusion of carbonaceous species | yes |
| Aerosol dry deposition | Zhang et al. (2001) |
| Horizontal advection scheme | van Leer |
| Vertical advection scheme | upwind |
| Urban correction | off |
| Resuspension process | off |
| Deep convection | on |
| Lateral boundary conditions | from C-IFS or MACC |

**Table 2.** *CHIMERE* parameters common to all sensitivity tests



| Acronym | Bio. Emis. | Anth. Emis. | Downscaling | Lu database | Hor. Res. (lat. × lon.) | Layers | Mechanism | Runtime |
|---|---|---|---|---|---|---|---|---|
| CS10E* | MEGAN | EMEP | lu, popul, traffic, lsp | GlobCover | WRF: 12×12km, CH: $0.15° \times 0.15°$ | 10 | SAPRC | 436s |
| CM10E | " | " | " | " | " | " | Full Melchior | 437s |
| CS20E* | " | " | " | " | " | 20 | SAPRC | 928s |
| CM20E | " | " | " | " | " | " | Full Melchior | 947s |
| FS10E* | " | " | " | " | WRF: 4×4km, CH: $0.05° \times 0.04°$ | 10 | SAPRC | 1598s |
| FM10E | " | " | " | " | " | " | Full Melchior | 1633s |
| FS20E* | " | " | " | " | " | 20 | SAPRC | 3582s |
| FM20E* | " | " | " | " | " | " | Full Melchior | 3755s |
| CS10H | " | HTAP | lu | USGS | WRF: 12×12km, CH: $0.15° \times 0.15°$ | 10 | SAPRC | not saved |
| CM10H | " | " | " | " | " | " | Full Melchior | " |
| CS20H | " | " | " | " | " | 20 | SAPRC | " |
| CM20H | " | " | " | " | " | " | Full Melchior | " |
| FS10H | " | " | " | " | WRF: 4×4km, CH: $0.05° \times 0.04°$ | 10 | SAPRC | " |
| FM10H | " | " | " | " | " | " | Full Melchior | " |
| FS20H | " | " | " | " | " | 20 | SAPRC | " |
| FM20H | " | " | " | " | " | " | Full Melchior | " |
| FM20H-P | " | " | lu, popul | " | " | " | " | " |
| FM20H-C | " | " | lu | " | WRF: 12×12km, CH: $0.05° \times 0.04°$ | " | " | " |
| FM20H-N | None | " | " | " | WRF: 4×4km, CH: $0.05° \times 0.04°$ | " | " | " |

**Table 3.** Overview of the applied sensitivity tests, C = coarse horizontal resolution, F = fine horizontal resolution, 10 = number of vertical layers, S = SAPRC, M = full Melchior, E = EMEP, H = HTAP, P = population downscaling, C = coarse meteorology, N = no biogenic emissions, lu = landuse, popul = population, lsp = emission allocation according to large point sources, Runtime in seconds for a typical summertime heat day (August 5th, 2018)