# Peer review of "An exploratory performance assessment of the *CHIMERE* model (version 2017r4) for the northwestern Iberian Peninsula and the summer season"

_Geoscientific Model Development, 2020_

## Referee Comment (RC1) · Anonymous Referee #1 · 13 Mar 2020

I already performed a review of the paper at ACP(D): https://www.atmos-chem-phys-discuss.net/acp-2019-351/

All relevant points from this first review were considered in the new submission. The quality of the manuscript increased clearly and it is well suited for publication in GMD(D). However, before I can recommend the manuscript for publication in GMD some minor issues need to be fixed:

(1) The authors performed a lot of simulations and when looking at the Figures (e.g.

Fig. 10) I feel totally lost. There are so many abbreviations which I simply cannot remember ( I know that Table 3 lists all of them). Of course the naming of simulations is always a discussion point and I see where it is problematic to find names which everyone 'likes'. However, I would suggest that the authors spend some lines of texts about their naming convention of their simulations so that the reader gets familiar with your naming convention.

(2) Figures 2-3 and 6-7 show statistical measures over the sub figures (SBIAS etc.). The text is simply to small for reading. Either put the numbers in separate tables, increase the size or delete them.

Technical comments: Please run a full spell checking of the revised manuscript. While reading I found some errors which should be found by a spell checker.

p3l71: horzizontal

p3l74ff: The sentence is unclear. It is also unclear why your cite Pay et al 2019 (they used CMAQ)

p9l24: high hores?

p17l525: troposfere

---

## Referee Comment (RC2) · Anonymous Referee #2 · 19 Apr 2020

As a reviewer I already examined and commented a previous version of the manuscript, which was initially submitted to ACP. The paper has been in accordance with editor and reviewers transferred to GMD, since it has a quite technical focus. With this newly submitted manuscript the authors addressed all critical points raised by me, I am fully satisfied with their response. The manuscript has considerably improved. The study has been extended by using additionally a more recent emission data set from EMEP, what gave the comparison with observations on a sound basis. Also at several passages text segments have been added which help the reader to better grasp how the

study was actually performed. It would have been interesting to also explore the applied pre-set functions used for the time disaggregation of the annual emissions for study region. This might be a task for a follow up analysis. There still are a few typos in manuscript and some numbers in the figure may be too small to be readable. Thus, standard editorial work is need. In conclusion, I recommend this manuscript for publication in GMD.

―――――――――――――――――

---

## Author Response (AR1)

**Author Response to the Editor**

Dear Dr. Jöckel,

Thank you very much for supervising the review process of our manuscript. Following your advise from our e-mail correspondence, we have corrected all the issues raised by the executive editor.

Your comment: *CHIMERE and WRF source code is on project websites rather than public persistent archives.*
Response: The CHIMERE and WRF project websites are mentioned in the "Code availability" section of the the revised manuscript.

Your comment: *Dropbox is not a public persistent archive. Your data is only 73 MB of data, so you could just put it on Zenodo.org.*
Response: We have uploaded our data to Zenodo.org and cite the corresponding doi in the "Data availability" section of the revised manuscript. Thanks for your advise.

Your comment: *The remaining data does appear to be in institutionally secure archives, but as a technical matter, the data should be precisely cited rather than just pasting URLs.*
Response: Whenever a doi is available for a given dataset it is cited in the "Data availability" section of the revised manuscript. Otherwise, we cite the corresponding study and provide the URL of the dataset. Thanks for pointing this out.

**Author Response to Anonymous Referee 1**

General Comment: *I already performed a review of the paper at ACP(D):*
*https://www.atmos-chem-physdiscuss.net/acp-2019-351/*
*All relevant points from this first review were considered in the new submission. The quality of the manuscript increased clearly and it is well suited for publication in GMD(D). However, before I can recommend the manuscript for publication in GMD some minor issues need to be fixed:*

Response: We would once again thank the referee very much for taking the time to review our manuscript and for his/her helpful comments. In the following, we provide a response list to the remaining minor issues.

Minor Comment 1: *The authors performed a lot of simulations and when looking at the Figures (e.g. Fig. 10) I feel totally lost. There are so many abbreviations which I simply cannot remember (I know that Table 3 lists all of them). Of course the naming of simulations is always a discussion point and I see where it is problematic to find names which everyone 'likes'. However, I would suggest that the authors spend some lines of text about the naming convention of their simulations so that the reader gets familiar with your naming convention.*
Response: In the revised manuscript, we have added a paragraph describing the rationale behind our abbreviations so that they can be grasped more easily by the reader (see lines 192-197). Thanks for pointing this out.

Minor Comment 2: *Figures 2-3 and 6-7 show statistical measures over the sub figures (SBIAS etc.). The text is simply to small for reading. Either put the numbers in separate tables, increase the size or delete them.*

Response: You are absolutely right, the text in the headers of these figures can be hardly read. Following your advise, we have removed these headers and now provide the respective results in an additional table (Table 4 of the revised manuscript).

Technical comment 1: *Please run a full spell checking of the revised manuscript. While reading I found some errors which should be found by a spell checker.*
Response: We have carefully corrected the manuscript with a spell checker, as suggested by you.

Technical comment 2: *p3 l71: horzizontal*
Response: This typo has been corrected in the revised manuscript.

Technical comment 4: *p3 l74ff: The sentence is unclear. It is also unclear why your cite Pay et al 2019 (they used CMAQ).*
Response: Here we mean that long-range transport events such as e.g. Saharan dust intrusions are not simulated by *CHIMERE* itself but are passed through from C-IFS at *CHIMERE's* lateral boundaries instead. In the revised manuscript, this is now more clearly pointed out in lines 73-77. Also, Pay et al. (2019) are no longer cited in this sentence.

Technical comment 2: *p9 l24: high hores?*
Response: This has been corrected in the revised manuscript.

Technical comment 3: P17 l525: *troposfere*
Response: This error has also been corrected.

**Author Response to Anonymous Referee 2**

Comment: *As a reviewer I already examined and commented a previous version of the manuscript, which was initially submitted to ACP. The paper has been in accordance with editor and reviewers transferred to GMD, since it has a quite technical focus. With this newly submitted manuscript the authors addressed all critical points raised by me, I am fully satisfied with their response. The manuscript has considerably improved. The study has been extended by using additionally a more recent emission data set from EMEP, what gave the comparison with observations a sound basis. Also at several passages text segments have been added which help the reader to better grasp how the study was actually performed. It would have been interesting to also explore the applied pre-set functions used for the time disaggregation of the annual emissions for study region. This might be a task for a follow up analysis. There still are a few typos in manuscript and some numbers in the figure may be too small to be readable. Thus, standard editorial work is need. In conclusion, I recommend this manuscript for publication in GMD.*

Response: We would like to thank the referee for taking the time to review our manuscript and for his/her helpful suggestions. For the future, we are planning to optimize the spatial and also the temporal disaggregation you mention for our region of interest in order to further improve the model system's performance. In the revised manuscript, the too small panel headers in Figures 2, 3, 6 and 7 have been removed and the corresponding results are now provided in an additional table (Table 4). We have also applied a spell checker on the latex file in order to correct all those typos that could not be detected by manual proofreading.

[revised manuscript text omitted]
 | -1.29 | 0.52 | 0.37 | 1.41 | -0.26 | -0.02 | 1.25 | 4.97 | 0.99 | 0.36 | 0.40 | 1.05 |

---

## Author Response (AR2)

**Author Response to the Editor**

Dear Dr. Jöckel,

Many thanks for supervising the review process of our manuscript. Following your minor comments, we have modified the Code and Data Availability sections as discussed in our e-mail correspondence. We have also added the configuration files for *all* of our experiments to our Zenodo entry and now cite the DOI of the WRF repository as suggested by the WRF developers.

As you can see in the latex difference file between version 3 and 4 of the manuscript, we have slightly modified one sentence in the abstract, included two additional references in line 155 and extended the Acknowledgements. We regret that our latexdiff version was not able to mark differences located in the Code and Data Availability sections.

With kind regards,

Swen Brands
On behalf of the author team

[revised manuscript text omitted]
 | -1.29 | 0.52 | 0.37 | 1.41 | -0.26 | -0.02 | 1.25 | 4.97 | 0.99 | 0.36 | 0.40 | 1.05 |